# Structure of antiviral drug bulevirtide bound to hepatitis B and D virus receptor protein NTCP

Hongtao Liu [1], Dariusz Zakrzewicz[2], Kamil Nosol [1], Rossitza N. Irobalieva [1], Somnath Mukherjee [3], Rose Bang-Sørensen [1], Nora Goldmann[4,5], Sebastian Kunz [2], Lorenzo Rossi [1], Anthony A. Kossiakoff [3] ✉, Stephan Urban [6,7] ✉, Dieter Glebe [4,5] ✉, Joachim Geyer [2] ✉ & Kaspar P. Locher [1] ✉

Cellular entry of the hepatitis B and D viruses (HBV/HDV) requires binding of the viral surface polypeptide preS1 to the hepatobiliary transporter Na⁺-taurocholate co-transporting polypeptide (NTCP). This interaction can be blocked by bulevirtide (BLV, formerly Myrcludex B), a preS1 derivative and approved drug for treating HDV infection. Here, to elucidate the basis of this inhibitory function, we determined a cryo-EM structure of BLV-bound human NTCP. BLV forms two domains, a plug lodged in the bile salt transport tunnel of NTCP and a string that covers the receptor's extracellular surface. The N-terminally attached myristoyl group of BLV interacts with the lipid-exposed surface of NTCP. Our structure reveals how BLV inhibits bile salt transport, rationalizes NTCP mutations that decrease the risk of HBV/HDV infection, and provides a basis for understanding the host specificity of HBV/HDV. Our results provide opportunities for structure-guided development of inhibitors that target HBV/HDV docking to NTCP.

With approximately 1.5 million individuals infected annually, the hepatitis B virus (HBV) continues to pose a substantial global health challenge, despite the availability of an effective vaccine[1]. HBV, a small, enveloped DNA virus, causes acute and chronic infection of the liver and the chronic form, in particular, significantly contributes to the overall burden of liver-related diseases, such as cirrhosis and hepatocellular carcinoma (HCC)[2,3]. Additionally, co-infection of HBV-infected patients with the hepatitis D virus (HDV), an enveloped RNA satellite virus that uses HBV surface proteins in HBV/HDV co-infected cells for envelopment and infection, enhances the severity of liver disease[4–6].

Since HBV and HDV share the HBV surface proteins to target hepatocytes for infection, their entry mechanisms have been studied extensively[7,8]. Viral entry involves initial low-affinity binding to cellular heparan sulfate proteoglycans (HSPGs)[9–12] followed by high-affinity binding to the human Na⁺-taurocholate co-transporting polypeptide (NTCP), an interaction crucial for HBV/HDV infection (Fig. 1)[8,13,14]. NTCP (SLC10A1), a member of the solute carrier transporter family, is one of four transporters involved in the enterohepatic circulation of bile salts[15–17]. Located in the basolateral membrane of hepatocytes, NTCP plays a vital role in transporting approximately 80% of bile salts,

[1]Institute of Molecular Biology and Biophysics, ETH Zürich, Zürich, Switzerland. [2]Institute of Pharmacology and Toxicology, Faculty of Veterinary Medicine, Justus Liebig University Giessen, Giessen, Germany. [3]Department of Biochemistry and Molecular Biology, The University of Chicago, Chicago, IL, USA. [4]Institute of Medical Virology, National Reference Centre for Hepatitis B Viruses and Hepatitis D Viruses, Justus Liebig University Giessen, Giessen, Germany. [5]German Center for Infection Research (DZIF) - Giessen-Marburg-Langen Partner Site, Giessen, Germany. [6]Department of Infectious Diseases, Molecular Virology, Heidelberg University, Heidelberg, Germany. [7]German Center for Infection Research (DZIF) - partner site Heidelberg, Heidelberg, Germany. ✉e-mail: koss@bsd.uchicago.edu; Stephan.Urban@med.uni-heidelberg.de; Dieter.Glebe@viro.med.uni-giessen.de; Joachim.M.Geyer@vetmed.uni-giessen.de; locher@mol.biol.ethz.ch

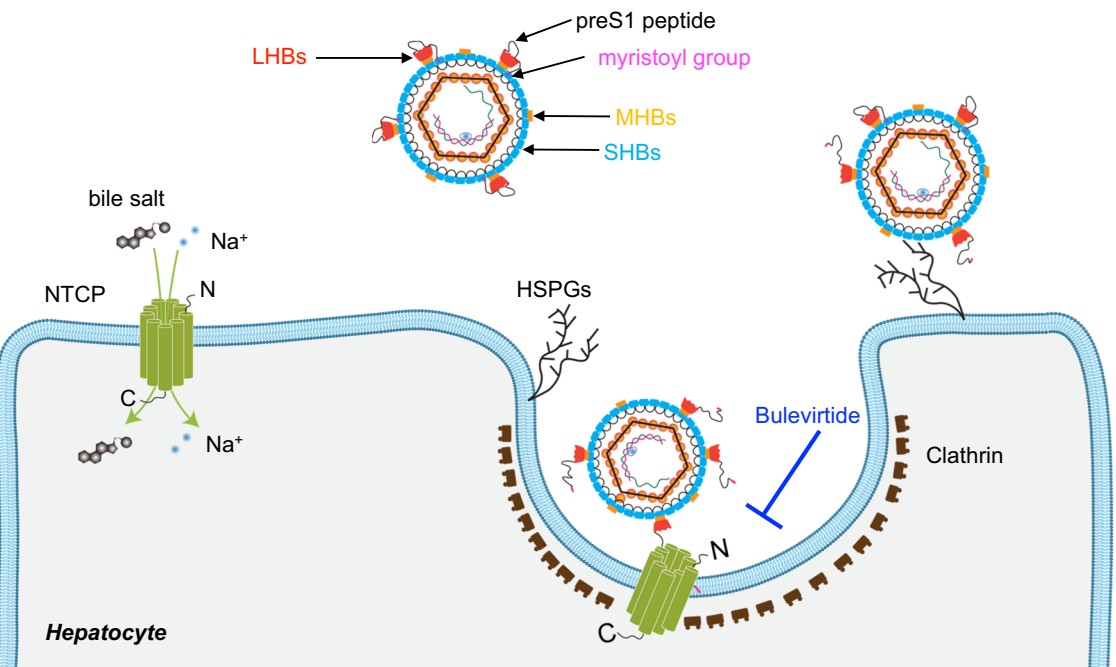

**Fig. 1 | Schematic of human NTCP-mediated HBV infection.** Bile salt uptake into hepatocytes is mediated by NTCP, located in the basolateral membrane of hepatocytes. HBV interacts with heparan sulfate proteoglycans (HSPGs) by low-affinity attachment, followed by interaction with the NTCP receptor protein, initiating the entry of viral particles into the hepatocytes. HBV-NTCP interaction occurs via high-affinity binding of the myristoylated preS1 domain of LHBs to NTCP. This process can be inhibited by bulevirtide, a preS1-derived peptide.

particularly conjugated bile salts, from the bloodstream into hepatocytes[18,19]. Its function is linked to maintaining bile salt homeostasis, essential for the digestion and absorption of dietary fats[18]. In 2012, NTCP was identified as a receptor for HBV/HDV[8,13,14]. In 2022, four independent studies reported structures of NTCP, revealing nine transmembrane helices (TMs) arranged in two domains, termed panel and core[20–23]. Two of these helices create a crossing motif (X-motif), near which two sodium ion binding sites are located[23]. The highest-resolution structure revealed the binding sites for two bile salt molecules in a continuous tunnel that links the extracellular and cytoplasmic side of the basolateral membrane[23].

The viral envelopes of HBV and HDV contain three surface proteins: large (LHBs), middle (MHBs), and small (SHBs), generated from a single open reading frame through alternative start codons[24–26]. They contain the same C-terminus but differ by N-terminal additions and N-glycosylation status. The SHBs contain four transmembrane helices that are embedded in the viral membrane and are linked by intra- and intermolecular disulfide bridges. The MHBs only contain a preS2-domain, whereas the LHBs contain a preS2- and a preS1-domain located N-terminal to the SHBs[27]. During protein synthesis, the LHBs undergo a posttranslational modification via myristoylation at a conserved glycine residue (Gly2) in the preS1-domain[24,28,29]. Previous studies have shown that the first 75 residues of preS1 are important for viral infectivity of HBV/HDV[30,31]. Residues [9]NPLGFFP[15] are essential for infectivity and binding and are highly conserved within the ten known genotypes of HBV (GtA-GtJ)[32]. Previous studies suggest that residues Gly2-Asp48 bind to NTCP[32]. Specific regions of NTCP, including extracellular loop 1 (ECL1) that connects TM2 and TM3 (residues Arg84-Asn87), and the extracellular surface of TM5 (residues Lys157-Leu165), have been shown to interact with the preS1 peptide[8,14,33]. The exact mechanism of NTCP-mediated virus entry remains elusive; however, it is thought to occur via endocytosis, which is known to recruit various host factors for entry initiation[34,35]. During active HBV infections, two distinct species of viral particles are produced: infectious virus particles and a far larger excess of non-infectious subviral particles (SVPs)[36–38]. Whilst both species contain all three viral envelope proteins, allowing for recognition and binding to NTCP as a receptor, SVPs lack genetic material and are therefore not infectious[38,39]. Thus, SVPs provide an alternative model system for the study of HBV/HDV receptor interaction.

It was recognized that strategies aimed at inhibiting or disrupting binding of preS1 to NTCP hold promise in preventing HBV/HDV infection. In 2023, the commercially available drug Hepcludex® (also known as bulevirtide (BLV), formerly Myrcludex B), received market approval for the treatment of chronic HDV infection in Europe[32,40–43]. The sequence of BLV is derived from that of preS1 from HBV genotype C (residues Gly2-Gly48) with a shortening of the 11 additional N-terminal amino acids and one amino acid substitution, Gln46Lys[32]. BLV exhibits a remarkably high inhibitory constant ($IC_{50} = 140$ pM) against HBV and HDV in primary human hepatocytes and HepaRG cells[32,44]. BLV has also been demonstrated to inhibit the NTCP-mediated uptake of bile salts, but at an $IC_{50}$ in the nanomolar range[45].

We here report a cryo-electron microscopy (cryo-EM) structure of BLV-bound human NTCP, simultaneously providing insight into the mechanism of BLV inhibition and into the interaction between NTCP and the viral preS1 peptide. Coupled with functional analysis, our study advances the molecular understanding of how BLV blocks HBV/HDV infection. These findings hold promise for developing additional therapeutic interventions against HBV/HDV, preventing viral entry into hepatocytes, and thus reducing HBV/HDV-related liver damage. Furthermore, our data may help rationalize why HBV infectivity is affected by variations in NTCP among different species.

## Results and discussion
### Interaction of bile salts, preS1, and patient-derived SVPs with NTCP
NTCP was shown to transport bile salts in a strictly sodium-dependent manner[46] and we also observed that a fluorescently labeled

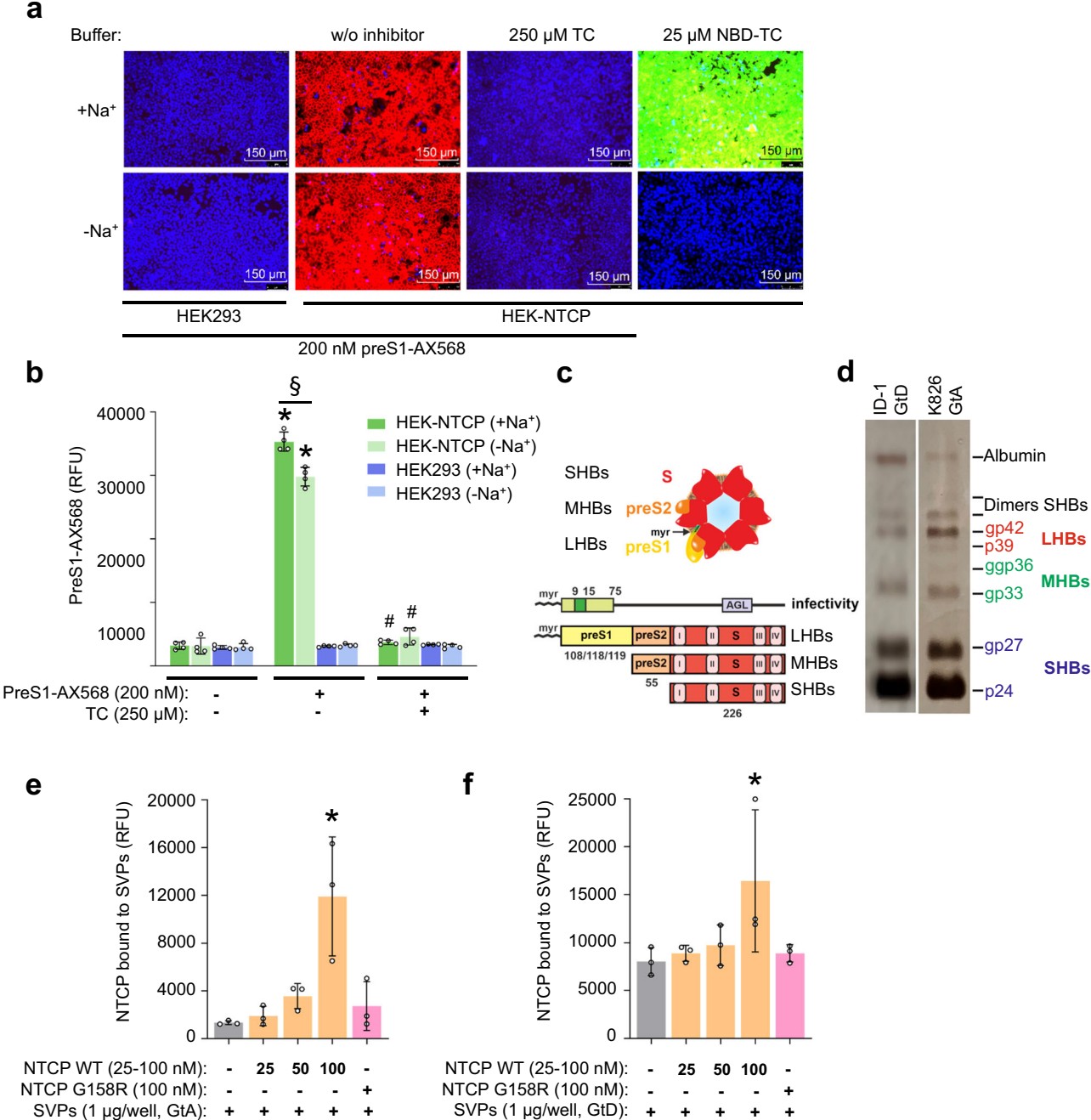

**Fig. 2 | Binding of preS1 and patient-derived SVPs as model systems to study HBV-NTCP interaction. a** Fluorescence microscopy and (**b**) fluorescence detection of HEK-NTCP cells and non-NTCP expressing HEK293 cells, incubated with or without 200 nM fluorescent preS1-AX568 peptide (representing genotype D, GtD) in the presence and absence of sodium. Expression levels of NTCP in HEK-NTCP cells were confirmed with qPCR. NBD-TC was used to demonstrate NTCP-mediated bile salt transport, whereas 250 µM TC was used as an inhibitor of preS1-peptide binding. Data in (**b**) present the means ± SD of quadruplicate measurements. *Significantly higher preS1-AX568 fluorescence compared to all other columns, §significantly different preS1 binding and #significant preS1 binding inhibition, all $p < 0.0001$. **c** Schematic of HBV subviral particles. The hepatitis B virus surface proteins LHBs, MHBs, and SHBs differ in the N-terminal additions (preS1, preS2) and N-glycosylation pattern within the S-domain. The antigenic loop (AGL) within the S-domain and the N-terminal preS1, together with the myristic acid covalently attached to Gly2, are determinants for HBV infectivity. **d** Silver-stained

polyacrylamide gels of highly purified SVP preparations from two different patients: K826 HBV-GtA and ID1 HBV-GtD. Both show clear bands of non- and N-glycosylated SHBs (p24/gp27 kDa), single and double N-glycosylated MHBs (gp33/ggp36 kDa), and non- and N-glycosylated preS1-containing LHBs (p39/gp42 kDa). Non-reduced SHBs-dimers (48 kDa or 54 kDa, depending on glycosylation status) and human serum albumin (67 kDa) are also indicated. **e, f** Binding of nanodisc-reconstituted wild-type or G158R mutant of NTCP-eYFP to SVPs from (**e**) patient K826 (HBV GtA) and (**f**) patient ID1 (HBV GtD). NTCP-eYFP fluorescence was detected using a fluorescence microtiter plate reader at 485 nm excitation and 535 nm emission. Data present the means ± SD of triplicate measurements of background-subtracted relative fluorescence units (RFU). *Significantly higher fluorescence intensity between SVP-coated wells incubated with and without nanodisc-reconstituted NTCP-eYFP ($p = 0.0013$, and $p = 0.0492$ for panels (**e**) and (**f**), respectively, according to one-way ANOVA with Dunnett's multiple comparison test).

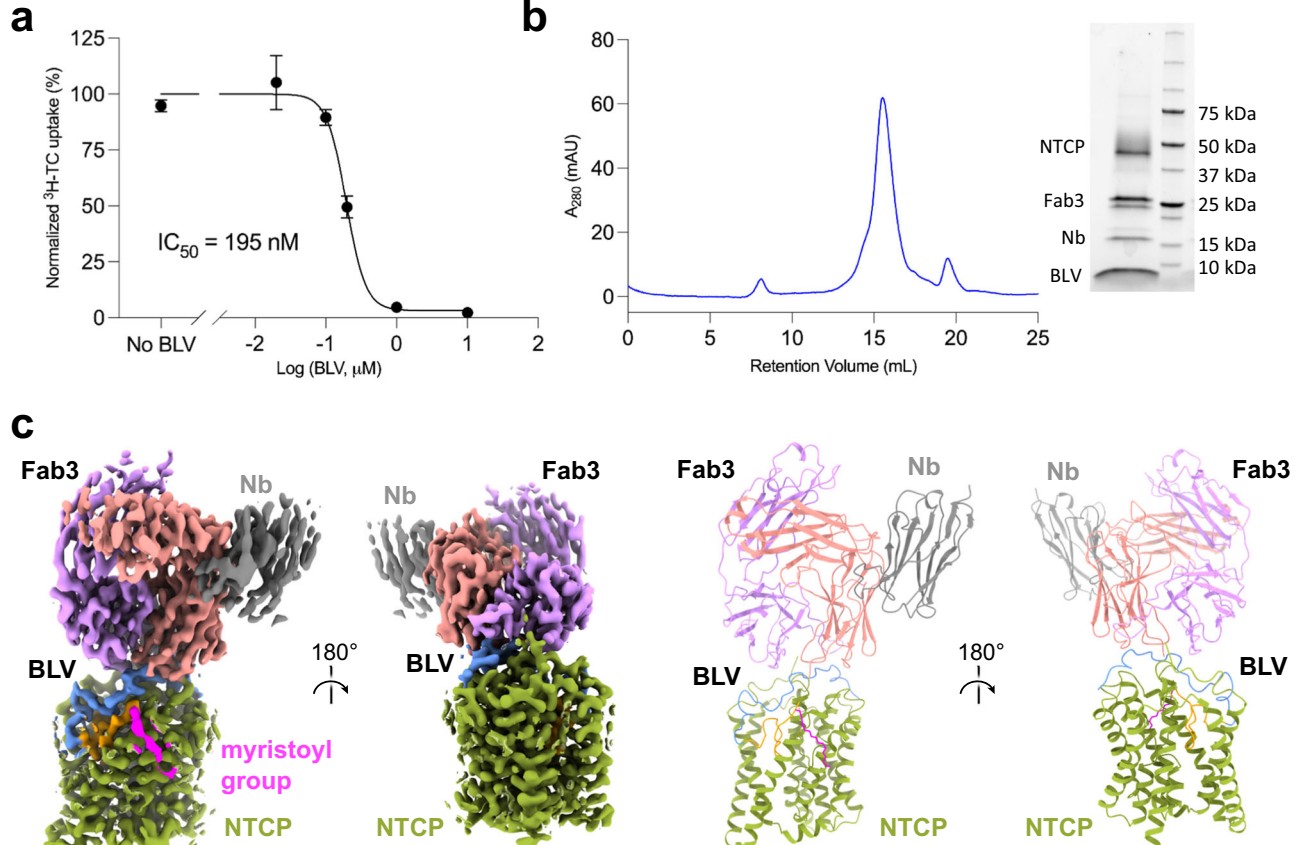

**Fig. 3 | Cryo-EM structure and functional analysis of BLV-bound NTCP.**
**a** Normalized inhibition of TC transport (1 μM) into Flp-In T-Rex cells stably expressing NTCP at varying concentrations of BLV. Uptake of $^3$H-TC was measured after 8 min, each data point indicates the mean of three independent replicates, and error bars represent the SD. **b** Size exclusion chromatography and SDS-PAGE analysis of purified human NTCP in a complex with BLV, Fab3, and Nb. **c** Cryo-EM density map (left) and model in ribbon representation (right) of NTCP-BLV-Fab3-Nb complex. The BLV peptide is colored in three sections: myristoylated glycine (magenta), plug (orange), and string (blue). The densities and models are colored as follows: NTCP in green, Nb in gray and Fab3 in purple (heavy chain) and coral (light chain).

taurocholate derivative (4-nitrobenzo-2-oxa-1,3-diazole-taurocholic acid, NBD-TC) is taken up into NTCP-expressing HEK293 cells in a sodium-dependent manner (Fig. 2a). Transport of NBD-TC can only be detected by using a transport buffer containing 143 mM NaCl, while NBD-TC fluorescence is undetectable in sodium-free buffer (equimolar substitution of NaCl with choline chloride). In contrast, the fluorescently labeled preS1 derivative preS1-AX568 binds to NTCP-expressing HEK293 cells under both sodium-containing and sodium-free conditions, indicating that preS1-binding to NTCP does not require sodium (Fig. 2b). Binding of preS1-AX568 to NTCP can be blocked by pre-incubation with 250 μM taurocholic acid (TC) (Fig. 2a, b), confirming previous data in NTCP-expressing HepG2 hepatoma cells[13]. We then analyzed if preparations of HBV SVPs obtained from patients chronically infected with HBV are capable of binding recombinantly expressed NTCP protein, as used for structure determination in the present study. HBV SVPs contain all surface proteins, namely SHBs, MHBs, and the preS1-bearing LHBs that are also enveloping the HBV/HDV virus particles (Fig. 2c). The SVP preparations represent the HBV genotypes D (GtD, patient ID1) and A (GtA, patient K826) (Fig. 2d). In both preparations the preS1-containing LHBs can be clearly detected by silver-stained polyacrylamide gel electrophoresis, although patient K826 showed a more prominent signal than patient ID1 (Fig. 2d). Plates were then coated with these patient-derived SVPs and binding of nanodisc-reconstituted, eYFP-labeled NTCP[23] was observed by fluorescence measurement. We found a concentration-dependent binding of NTCP (25–100 nM) to GtA and GtD SVPs, demonstrating binding of NTCP to

the SVPs. As a control, we used an NTCP variant carrying the mutation G158R, which abolished binding to the SVPs (Fig. 2e, f).

### Functional analysis and BLV on TC transport
Bile salt transport in NTCP-expressing HEK293 cells was also analyzed with the radiolabeled bile acid $^3$H-TC that showed robust uptake into the cells in the presence of media containing sodium (Fig. 3a). TC transport was inhibited at increasing concentrations of BLV with an inhibition constant value (IC$_{50}$) of 195 nM (95% CI, 174–218 nM) (Fig. 3a). At high concentrations, BLV fully abolished transport. This is consistent with the previously shown result for myristoylated preS1 inhibition of TC uptake into NTCP-expressing HEK293 cells (IC$_{50}$ of 190 nM)[13,47], indicating that our NTCP construct is functional and suitable for further studies on interaction with BLV.

### Isolation of a Fab specific to the NTCP-BLV complex
Several antibody fragments specific to human NTCP were previously reported[20–23]. The preS1 peptide is known to interact with the extracellular surface of NTCP[8,14]. All binders reported in the literature bind to the same region[20–23], suggesting an overlap in binding epitopes and therefore, incompatibility with preS1 and BLV. Hence, to gain structural insight into how BLV interacts with NTCP we aimed to generate an antigen-binding antibody fragment (Fab) specific to the NTCP-BLV complex. We generated a complex of detergent-solubilized, biotinylated NTCP and BLV (Supplementary Fig. 1a), which was used as a target for isolating Fabs from synthetic library E[48] using phage display

(Supplementary Fig. 1b). During each round of the selection, a molar excess of the BLV peptide was maintained to ensure saturation of the binding epitope. We identified one Fab, termed Fab3, that formed a stable complex with NTCP in the presence of BLV, with an apparent $K_d$ value of 8 nM (Supplementary Fig. 2). We found that Fab3 could not form a complex with NTCP in the absence of BLV (Supplementary Fig. 1c), suggesting that Fab3 specifically recognizes the NTCP-BLV complex.

## Cryo-EM structure of human NTCP bound to BLV

For structural studies, we formed a complex of NTCP (38 kDa), BLV, Fab3 (50 kDa) and a Fab-binding nanobody[49] (Nb, 15 kDa), and used a monodisperse fraction of the sample for single particle cryo-EM analysis (Fig. 3b). We obtained a 3.4 Å EM density map revealing excellent density for NTCP, bound BLV, and Fab3. The structure of NTCP was similar to that of the previously reported, substrate-bound NTCP[23], featuring a translocation tunnel in an open conformation (Fig. 3c and Supplementary Fig. 3). The EM map revealed well-resolved density for BLV located both in the tunnel and on the surface of NTCP, allowing us to build all 47 amino acids and the myristoyl moiety of the BLV peptide (Supplementary Fig. 4). Consistent with our biochemical analysis, Fab3 is bound on the extracellular surface and interacts with the C-terminus residues of BLV and four regions of NTCP (N-terminus, ECL1, ECL2, and ECL4) (Fig. 3c).

## BLV interactions with NTCP and Fab3

BLV has a large interface with NTCP, with 2064 Å$^2$ buried surface area between them, rationalizing the tight binding of BLV (and by extension preS1) to its receptor. Based on the structural observations, we divided the sequence of the BLV peptide into three sections - the myristoyl group, the plug (residues Gly2-Asp20), and the string (residues Pro21-Gly48) (Fig. 4a, b). Note that for consistency, residue numbering of BLV follows that of the preS1 domain of LHBs of genotype D (HBV GtD) (Fig. 4c). The myristoyl group of BLV interacts with the surface of TM4 (Phe128, Leu131, and Met133) and TM5 (Tyr156) of NTCP and is exposed to lipids from the outer leaflet of the basolateral hepatocyte membrane. Lipidation has been identified as a crucial factor in stabilizing the binding of viral peptides[32], and a previous study demonstrated a significant decrease in viral peptide binding in the absence of myristoylation[32]. Absence of N-terminal myristoylation of preS1 by experimental mutation of Gly2 did not lead to abrogation of viral assembly and release but rendered HBV non-infectious[50,51]. Furthermore, during the development of BLV, peptides that did not contain a myristoyl moiety exhibited low or negligible antiviral activity[32]. The location of the myristoyl group places the first amino acid, Gly2, at the external entrance of the NTCP tunnel (Fig. 4a, d). Interestingly, for most HBV genotypes (besides genotypes D and J), the preS1 peptide contains up to an additional 11 residues between the myristylation motif and Gly2 of the plug domain (Fig. 4c). Our structure suggests that these residues are not involved in receptor recognition as they are most likely exposed to the cell exterior and would not disturb the interaction of the plug and string domains of preS1 with NTCP.

Starting with Gly2, the first 19 residues of BLV adopt a globular shape that forms a plug wedged inside the translocation tunnel and reaching the middle of the membrane bilayer (Fig. 4d, e). It was previously shown that deleting the plug region from inhibitory myr-preS1 peptides (HBVpreS/19-48$^{myr}$) abrogates their inhibitory potential on HBV infection[28,40]. In addition, deletion within the preS1 plug region of LHBs abolishes the infectivity of HBV[31]. Notably, residues Gly2-Asp20 are highly conserved (Fig. 4c). Only three of the plug residues, Thr3, Val7, and Pro8, differ among the preS1 genotypes: Thr3 is exposed to the solvent, whereas Val7 and Pro8 are replaced in preS1 genotypes D, E and G by ones of a similar size, such as threonine, serine, or alanine. Furthermore, several conserved residues of the BLV peptide (Leu11-

Phe14) contain hydrophobic side chains exposed toward the lumen of the translocation tunnel (Fig. 4e). Among these, Phe14 is replaced by a leucine residue only in genotypes G and I. When the mutation is introduced in genotype C (as in BLV), it does not affect the ability of the virus to bind to hepatocytes[8].

The string domain of BLV (Pro21-Gly48) covers the surface of the plug (Fig. 4f) and crosses the extracellular surface of NTCP (Fig. 4g). Its importance for viral infectivity was previously established[31]. Deletions in infection-interfering myr-preS1 peptides (Δ20-21, Δ20-23, Δ23-27) reduce, but do not abolish the inhibitory activity of those peptides in competition assays with HBV[28]. Several side chains of the string domain are exposed to the solvent (Asn28, Asn29, Asp31, Asn37, Glu43, Asn45, Lys46) and do not interact with NTCP, which rationalizes why some of these are poorly conserved within preS1 among different genotypes (Fig. 4c). The C-terminus of the string (Glu42-Gly48) binds to ECL1 of NTCP, a region that contains the sequence $^{84}$RLKN$^{87}$ reported to be essential for host discrimination specificity[8,14,33]. This region is variable in different mammalian species. In mice, the corresponding residues are $^{84}$HLTS$^{87}$, and it was observed that mice hepatocytes can bind human HBV/HDV but cannot be infected. However, transgenic mice containing a humanized fragment of residues 84–87 in their NTCP can be infected[14,33,52]. While our structure can rationalize why mouse NTCP can bind preS1 of human HBV, it does not offer a direct reason why infectivity is unable to proceed, which is likely due to downstream events.

## Binding of BLV to NTCP prevents bile salt transport

We superimposed the NTCP-BLV structure with our previously reported, substrate-bound NTCP structure containing two molecules of glycochenodeoxycholic acid (GCDC, PDB: 7ZYI). While the overall root-mean-square deviation (RMSD) was only 0.767 Å, the structural differences were unevenly distributed (Fig. 5). Although the core domains of the two structures are virtually identical, the helices TM1, TM5, TM6 forming the panel domain are shifted, which is likely a consequence of BLV binding. The superposition shows that the plug of the BLV peptide partly overlaps with the substrate binding pocket. Simultaneous binding is therefore impossible because it would result in a steric clash (Fig. 5). Residues Leu11 and Phe13 of BLV overlap with the scaffold of GCDC, indicating that these side chains would prevent the binding of even the shortest bile acids (Fig. 5b). Our structure therefore rationalizes why BLV (and by extension preS1) binding to NTCP is incompatible with the binding and transport of bile salts. For patients administered BLV, an increase in bile salt levels in the plasma was observed. However, a recent clinical study showed that this increase is not significant enough to cause cholestatic liver injury or hepatocyte dysfunction[53].

In our previously reported structure of substrate-bound NTCP, we identified two sodium binding sites in the vicinity of the X-motif (Supplementary Fig. 5)[23]. However, in the BLV-bound NTCP structure, there is no density for sodium ions and the residues that have previously been shown to coordinate the binding of sodium ions, such as Glu257 and Gln261, are poorly resolved. This observation suggests that the presence of the BLV peptide prevents sodium ions from binding at high affinity, which is in line with the experimental evidence that the preS1 peptide can bind NTCP also in the absence of sodium (Fig. 2a, b), whereas bile salt transport requires sodium (Fig. 2a).

## Structural implications for HBV infectivity

Given the similarity between BLV and preS1, our structure can explain the difference between variants of NTCP among species and their effect on HBV infectivity. We aligned the sequence of human NTCP to that of Cynomolgus monkey and Common squirrel monkey, which are representatives of the Old and New World monkeys, respectively (Fig. 6a and Supplementary Fig. 6). Old World monkeys are not susceptible to HBV infection due to the large side chain of arginine at

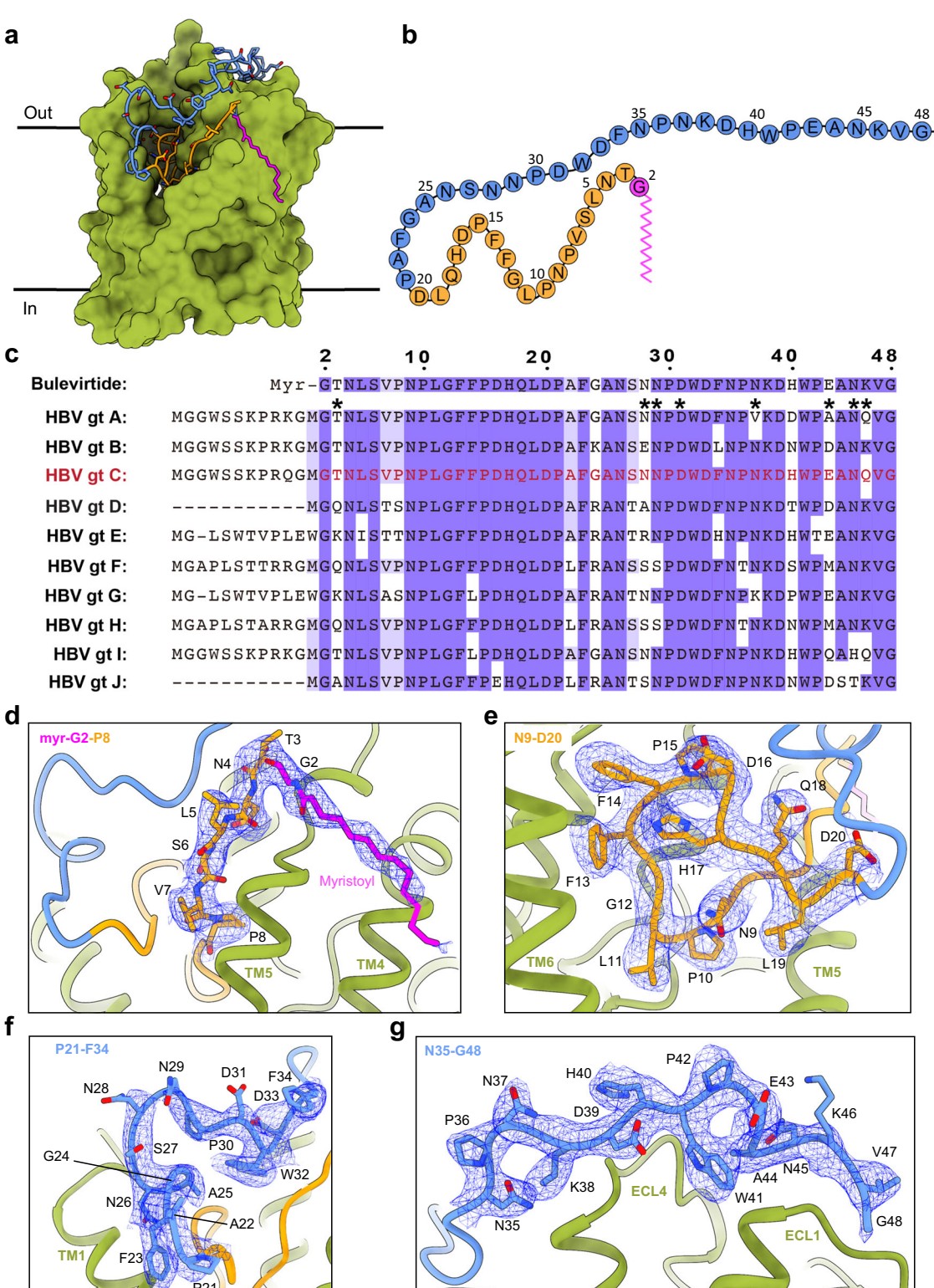

**Fig. 4 | Molecular interaction between BLV and NTCP. a** Surface representation of NTCP (green), with the bound BLV peptide: myristoylated glycine (magenta), residues T3-G20 (orange), and residues P21-G48 (blue). **b** 2D schematic of the full BLV peptide, coloring is the same as in (**a**). **c** Sequence alignment of BLV and the preS1 peptides from the ten genotypes of HBV. Purple shading indicates amino acids conservation between BLV and different HBV preS1 genotypes. Black asterisks denote residues of BLV that are exposed to the solvent and therefore do not interact with NTCP. Red font indicates the HBV genotype used for the design of the BLV peptide. **d**–**g** Interactions between BLV and NTCP. Coloring as in panel (**a**). The EM density for BLV is displayed as blue mesh. BLV residues are labeled black.

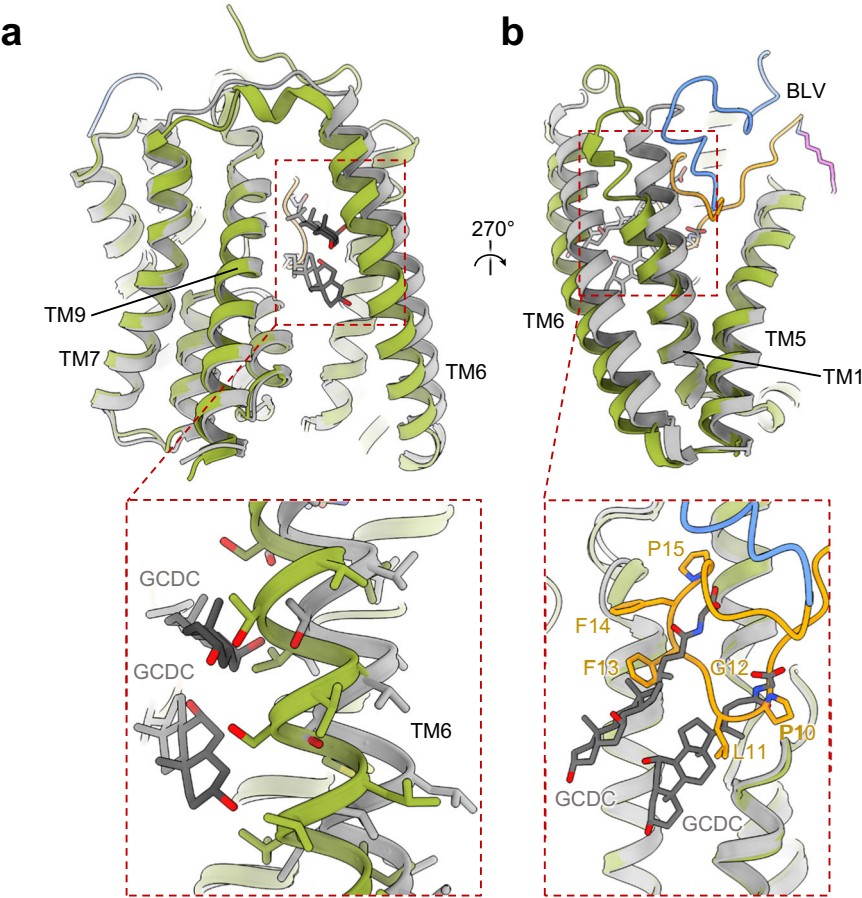

**Fig. 5 | Comparison of BLV-bound NTCP and substrate-bound NTCP.**
**a**, **b** Superposed BLV-bound NTCP (green) and substrate-bound NTCP (PDB ID: 7ZYI, gray) structures shown in ribbon representation. Two molecules of the substrate glycochenodeoxycholic acid (GCDC) are shown as gray sticks and the BLV peptide is colored as before. Close-up views show a clash between the substrate GCDC and TM6 of the BLV-bound NTCP structure (**a**), and a clash between GCDC substrates and the BLV peptide (**b**).

position 158 of NTCP[54]. Mutation of this Arg158 into Gly158, which is conserved in human and most of the New World monkey NTCPs, renders the Old World Monkey susceptible to human HBV in vitro[54]. Vice versa, Gly158Arg mutation of human and Common squirrel monkey NTCPs completely abolishes preS1 binding and HBV infection in the cell culture model, while maintaining the transport activity of NTCP for TC[54]. Our structure shows that Gly158 is located at the NTCP tunnel entrance, where it tightly packs against main chain atoms of the plug domain of BLV (Fig. 6b). As a result, any amino acid other than glycine at this position, and in particular a bulky amino acid such as arginine, will cause a steric clash with the plug of BLV (and by extension preS1), thereby preventing binding and HBV entry and infection (Fig. 6b). This is consistent with previous studies that showed that substitution of Gly158 interferes with preS1 binding[54–56].

Another noteworthy NTCP site is the nonsynonymous human genetic variant S267F (c.800C>T) of NTCP, which has been identified exclusively among East Asian populations, with a minor allele frequency ranging from 2% to 10%, while it remains absent in African and European populations[57]. The S267F variant cannot efficiently transport bile salts[58], which is in line with the position of Ser267 near the bile acid binding sites of NTCP[23,57]. In addition, the S267F mutation reduces HBV entry and infection in cell culture experiments and has been associated with resistance to chronic HBV infection and decelerated progression of related liver diseases[59,60]. Our structure visualizes the proximity of Ser267 to the plug region of the BLV peptide (Fig. 6c), where a phenylalanine would cause a steric clash

both with preS1 peptides, BLV and bile salts, which is in agreement with experimental findings[22].

While preparing this manuscript for submission, a structural study of NTCP bound to a preS1 peptide was published[61]. In their study, Asami et al. generated a NTCP-preS1 specific Fab that binds to a similar epitope as Fab3 presented in our study (Supplementary Fig. 7). Both structures reveal similar conformations of NTCP (RMSD = 0.496 Å) and the same fold for BLV (our study) and preS1 of genotype B[61]. This suggests that the Fab binders developed independently did not influence the fold of the peptides and their interactions with NTCP.

In conclusion, our study reveals the molecular basis of viral preS1 interactions with its receptor NTCP and the inhibition of this interaction by the commercial drug bulevirtide/Hepcludex® (Fig. 6d). The results also rationalize why binding of BLV affects bile salt transport by NTCP. Furthermore, our data corroborates previous biochemical results regarding which residues in the BLV sequence are critical for inhibiting viral infection. Our findings allow us to rationalize HBV/HDV specificity for human NTCP and explain why the S267F mutation in humans presents a compromised bile salt transport with simultaneous resistance to HBV/HDV. We identified BLV residues that face the solvent and do not appear to interact with NTCP. These might be amenable for the design of smaller drugs (peptidomimetics) that improve the pharmacology of BLV in order to prevent HBV and HDV binding and infection. Our structure also provides a starting point for rational design of HBV drugs that are not based on peptide backbones and may allow oral administration.

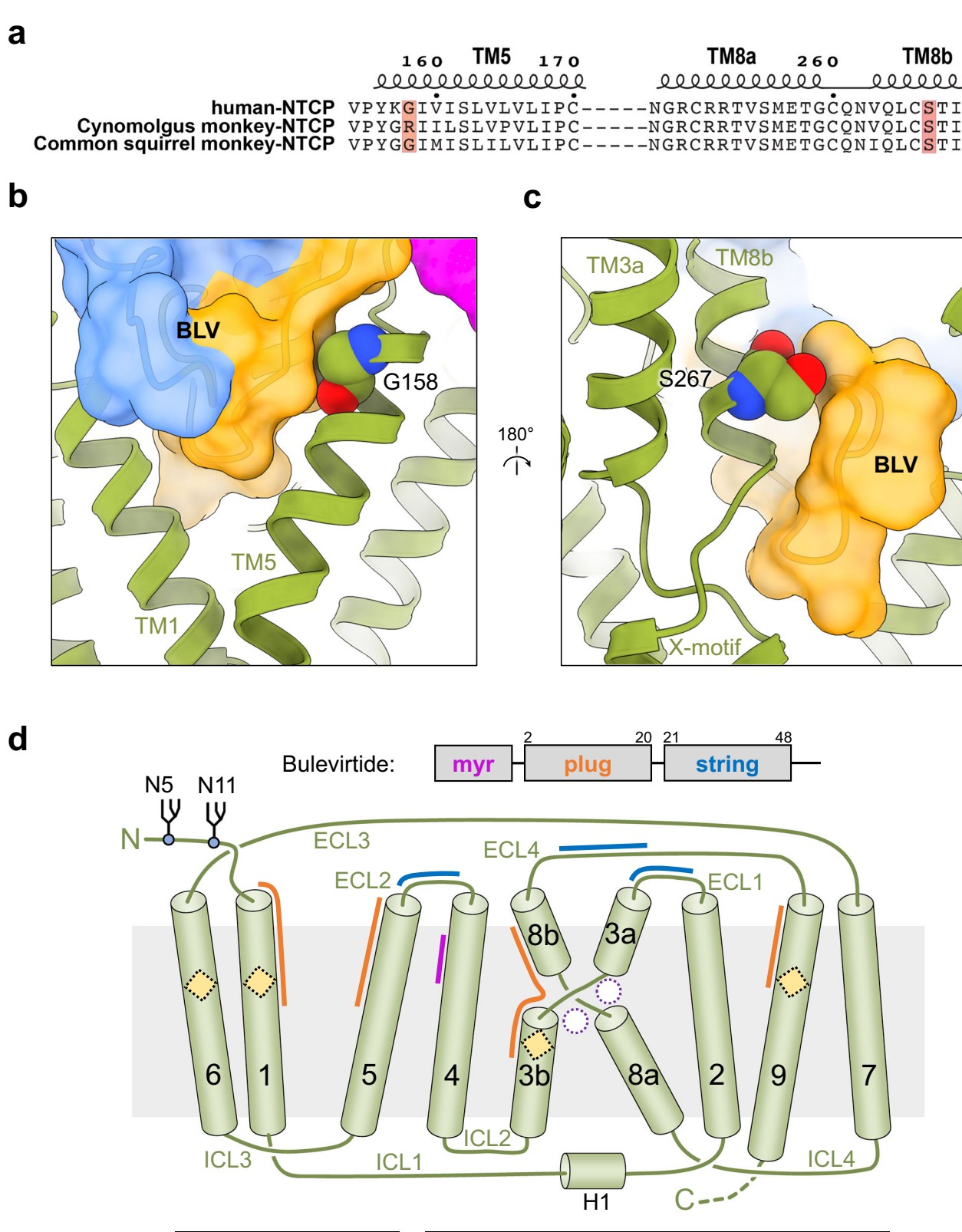

**a**

| | 160 | TM5 | 170 | | TM8a | 260 | TM8b |
| human-NTCP | VPYK**G**IVISLVLVLIPC | | ----NGRCRRTVSMETGCQNVQLC**S**TILN | | | | |
| Cynomolgus monkey-NTCP | VPYG**R**IILSLVPVLIPC | | ----NGRCRRTVSMETGCQNVQLC**S**TILN | | | | |
| Common squirrel monkey-NTCP | VPYG**G**IMISLILVLIPC | | ----NGRCRRTVSMETGCQNIQLC**S**TILN | | | | |

**b** BLV, G158, TM5, TM1

**c** 180° TM3a, TM8b, S267, BLV, X-motif

**d** Bulevirtide: myr | plug (2–20) | string (21–48)

N5 N11
N — ECL3, ECL2, ECL4, ECL1
8b, 3a, 3b, 8a, 2, 9, 7
ICL3, ICL1, ICL2, H1, ICL4, C
panel domain | core domain

## Methods

### Protein expression and purification

The full-length wild-type human NTCP gene (UniProt ID: Q14973) was generated synthetically by GeneArt (Thermo Fisher Scientific) after codon optimization for expression in HEK293 cells. All subsequent modifications of the sequence, including the introduction of point mutations, were performed using synthetic gene fragments and were confirmed via sequencing (Microsynth). NTCP was generated as a fusion construct, containing a C-terminal 3C protease cleavage site, followed by an eYFP-rho-1D4 tag. The NTCP construct used for Fab generation contained an Avi-tag between the C-terminal end of NTCP and the 3C protease cleavage site. Stable cell lines for expression were

**Fig. 6 | NTCP species specificity against HBV infection. a** Sequence alignment between human NTCP and NTCP from two different species of monkeys: Cynomolgus monkey (Old World) and Common squirrel monkey (New World). Highlighted amino acids indicate the residues described in the text: the Gly/Arg variation at position 158, and Ser267 and its genetic variant Phe267, which has been linked to HBV resistance. **b** A close-up view of residue Gly158 (sphere representation) and its proximity to BLV (orange and blue transparent surface). **c** A close-up view of residue Ser267 (sphere representation) and its proximity to BLV (orange and blue transparent surface). **d** Topology diagram of human NTCP. Sodium binding sites (unoccupied in the NTCP-BLV structure) and substrate binding regions are indicated as empty circles and yellow diamonds, respectively. ECL extracellular loop, ICL intracellular loop. The crossing motif (X-motif) is formed by TM helices 3 and 8. Glycosylation sites are shown and numbered. Colored arches indicate the regions of NTCP interacting with distinct regions of the BLV peptide.

---

generated using the doxycycline-inducible Flp-In T-Rex 293 system (Thermo Fisher Scientific, R78007) according to the manufacturer's guidelines.

Cells were adapted and maintained in the fresh complete Dulbecco's Modified Eagle Medium (DMEM, Gibco) supplemented with 10% fetal bovine serum (FBS, Thermo Fisher Scientific). Cells were grown at 37 °C under humidified conditions with 5% $CO_2$. Cell expression was induced by adding doxycycline (Sigma, D5207) to a final concentration of 3 µg/mL, whereafter cells were grown for a further 60 h in DMEM supplemented with 2% FBS under the previously mentioned conditions. Cells were harvested and flash-frozen in liquid nitrogen for storage at −80 °C. All purification steps were performed at 4 °C or on ice, whenever possible. Frozen cell pellets were thawed and homogenized with a Douncer in working buffer (10:1 vol/wt) containing 25 mM HEPES (pH 7.5), 150 mM NaCl, 20% glycerol, supplemented with cOmplete EDTA-free protease inhibitor tablets (Roche, 11873580001). Lauryl maltose neopentylglycol (L-MNG, Anatrace, NG310) was added to a final concentration of 1% (wt/vol). Solubilization took place for 2 h at 4 °C with gentle agitation before centrifugation at 140,000 × g for 30 min using a Type Ti-45 rotor (Beckmann). The supernatant was applied to Sepharose-coupled rho-1D4 antibody resin (University of British Columbia), previously equilibrated with 10 column volumes (CVs) of working buffer supplemented with 0.01% L-MNG. This mixture was incubated for 2 hours at 4 °C with gentle agitation. The 1D4 resin was hereafter washed three times with 10 CVs of washing buffer containing 25 mM HEPES (pH 7.5), 150 mM NaCl, 20% glycerol, and 0.01% L-MNG to remove unbound components. Subsequently, the 1D4 resin was incubated for 2 h with three CVs of buffer supplemented with a 1:50 (wt:wt) ratio of 3C protease to expected protein for cleavage of the C-terminal tags of the NTCP fusion protein. The elution was concentrated using a 50 kDa molecular weight cut-off centrifugal filter (Amicon) before loading onto a Superose 6 increase column 10/300 GL (Cytiva) for size exclusion chromatography. The protein concentration in detergent micelles was determined by measuring absorbance at 280 nm using a NanoDrop 2000c spectrophotometer (Thermo Fisher Scientific).

### Nanodisc reconstitution

A mix of brain polar extract lipids (Avanti Polar Lipids, 141101C) and cholesterol (Avanti Polar Lipids, 700100) (4:1 wt/wt) was solubilized in 1%/0.2% DDM/CHS (wt/wt), and thereafter sonicated. The lipids were then mixed with detergent-purified NTCP and incubated at room temperature for 5 min with gentle agitation. Next, membrane scaffold protein (MSP1D1) was added to the mixture and incubated for 20 min at room temperature. The molar ratio of the mixture was 1:5:100 (protein:MSP1D1:lipids). Bio-Beads SM-2 (Bio-Rad, 1528920) were activated with methanol, pre-equilibrated with HBS buffer (25 mM HEPES pH 7.5, 150 mM NaCl), and were added to a concentration of 0.8 g/mL to the nanodiscs mixture and incubated overnight at 4 °C with gentle mixing. Bio-Beads were removed by passing through a gravity column (Bio-Rad), and the mixture was briefly spun at 4000 × g at 4 °C to remove excess lipids. The sample was concentrated using a 50 kDa molecular weight cut-off centrifugal filter (Amicon), followed by size exclusion chromatography run as previously described, except now in HBS buffer. Peak fractions

containing nanodisc-reconstituted NTCP were collected and concentrated.

### PreS1-AX568 binding to HEK-NTCP cells

Human embryonic kidney Flp-In T-Rex 293 cells (Thermo Fisher Scientific, R78007, hereafter referred to as non-NTCP expressing HEK293 cells) stably expressing the human NTCP protein were generated according to the manufacturer's guidelines as reported before[62] and are hereafter referred to as HEK-NTCP cells. Functional NTCP expression was confirmed by NBD-TC transport[63]. Cells were maintained at 37 °C, 5% $CO_2$, and 95% humidity in Dulbecco's modified Eagle medium (DMEM)/Ham's F12 medium (Thermo Fisher Scientific, 41966029) supplemented with 10% fetal calf serum (Thermo Fisher Scientific, 10099141), 4 mM L-glutamine (PAA), and antibiotics (100 U/mL penicillin and 100 µg/mL streptomycin, both Anprotec) and 100 µg/mL hygromycin (Carl Roth). For induction of NTCP expression, HEK-NTCP cells were incubated with 1 µg/mL tetracycline (Carl Roth, HP63.1)[62]. Nuclei were stained with Hoechst33342 (Thermo Fisher Scientific, 62249) and fluorescence images were analyzed on a DMI6000 B inverted fluorescent microscope (Leica). Qualitative and quantitative preS1-peptide binding was analyzed with the NH2-terminally myristoylated and COOH-terminally AlexaFlour 568-coupled fluorescent myr-preS1[2-48] peptide (here referred to as preS1-AX568 peptide), consisting of amino acids 2-48 of the large HBV sub-genotype D3 surface protein (Biosynthesis). Briefly, HEK293 and HEK-NTCP cells were incubated with 200 nM preS1-AX568 peptide in sodium buffer (142.9 mM NaCl, 4.7 mM KCl, 1.2 mM $MgSO_4 \cdot 7H_2O$, 1.2 mM $KH_2PO_4$, 20 mM HEPES, 1.8 mM $CaCl_2 \cdot 2H_2O$, pH 7.4) or in sodium-free buffer (equimolar substitution of NaCl with choline chloride). Inhibitor pre-incubation was performed over 5 min at 37 °C, consistent with previously published data[64]. Quantification of relative fluorescence was assessed at 570 nm excitation and 615 nm emission on a fluorescence microplate reader (TECAN Life Sciences).

### Patient-derived SVP preparation and binding assay

Plasma samples from HBV-infected patients were obtained from fully anonymized persons and used in accordance with the ethics committee of the Justus Liebig University Giessen (AZ 257/18). Subviral particles (SVPs) were isolated from plasma of two chronic HBV-infected patients by three consecutive ultracentrifugation steps according to published protocols[13]. Briefly, SVP were separated from virions using rate zonal ultracentrifugation through a sucrose density gradient. SVP-containing fractions were pooled and further purified using caesium chloride (CsCl) flotation ultracentrifugation. Pooled SVP-fractions were further purified using a second sucrose density gradient and concentrated by ultrafiltration. SVP preparations from plasma of two different high-titer ($>10^9$ HBV genomes/mL), HBeAg- and HBsAg-positive, chronic HBV-infected patients of genotype A (patient #1, K826, HBV-GtA) and genotype D (patient #2, ID1, HBV-GtD) were used for binding assays with nanodisc-associated recombinant NTCP. Aliquots of the SVP preparations were analyzed by silver-stained polyacrylamide gels and revealed 6.3% and 12.03% LHBs for the patient #2 and #1 preparations, respectively. FluoroNunc polysorb F16 black plates (Nunc Thermo Scientific) were coated with SVPs (1 µg total/well, 20 µg/mL) overnight at 4 °C and afterward blocked with HBS buffer

(20 mM HEPES, pH 7.4, 150 mM NaCl) supplemented with 3% (w/v) bovine serum albumin (BSA) for 1 h at 4 °C under gentle shaking. After 3× washing with HBS buffer, 50 μL (25–100 nM) of nanodisc-reconstituted either wild-type or G158R NTCP was added onto the plate and incubated for 1 h at 4 °C under gentle shaking. The supernatant was slowly removed, and the samples were 3× gently washed with HBS buffer. For fluorescence measurement, 100 μL HBS buffer was added to each well and the fluorescence intensity of eYFP was detected at 485 nm excitation and 535 nm emission on a fluorescence microplate reader (TECAN Life Sciences).

### Transport assay into HEK cells

The uptake of radioactive taurocholate ($^3$H-TC, American Radiolabeled Chemicals) was used to determine the transport activity of NTCP. All reactions were performed in DMEM medium at 37 °C. The Flp-In T-Rex 293 cells expressing NTCP were induced upon the addition of 1 μg/mL tetracycline (AppliChem, A2228) and grown for 24 h under standard conditions. Hereafter, the cells were detached, and 150,000 cells were seeded into each well of a Poly-D-Lysine (Sigma, P6407) coated 24-well plate (Nunc, Thermo Fisher Scientific). After the cells had adhered, the media was exchanged with a pre-warmed buffer, consisting of DMEM, and where indicated, supplemented with a dilution of BLV dissolved in water, and pre-incubated for 15 min at 37 °C with gentle shaking. Hereafter, the media was removed and replaced with the pre-warmed uptake buffer, consisting of DMEM supplemented with 1 μM sodium taurocholate (TC) and BLV, maintaining the same BLV concentration as for the pre-incubation. The added TC contained a mix of radiolabeled TC ($^3$H-TC) and non-labeled TC in a 1:20 ratio (mol/mol). The uptake reaction was stopped after 8 min by washing the cells thrice with ice-cold phosphate-buffered saline (PBS, Gibco). Lysis buffer containing 1 M NaOH and 2% Triton X-100 in $H_2O$ was added, whereafter lysed cells were added to 2 mL scintillation fluid and radioactivity was measured using a scintillation counter (Perkin Elmer 2450 Microbeta2). Data were analyzed using GraphPad Prism 8.0.0 and fitted to a four-parameter logistic curve from where the $IC_{50}$ and the 95% confidence interval were calculated. The data were normalized to the top plateau of the curve.

### Enzymatic biotinylation of NTCP

We used biotinylated NTCP for Fab selection from the synthetic Fab library E[48]. 3C-cleaved NTCP in detergent was biotinylated via BirA-mediated biotinylation of the NTCP Avi-tag construct in the presence of 1 μM BLV, as previously described[23]. The efficiency of biotinylation was verified by a streptavidin pull-down assay.

### Phage display selection

Detergent-solubilized, biotinylated NTCP was used for phage display selection. The biotinylated Avi-tagged protein was in 40 mM HEPES, pH 7.4, 150 mM NaCl, and 0.01% L-MNG. Phage display selection was performed using the phage library E[48] following published protocols with necessary modifications[65]. 1 μM BLV peptide was used in selection to obtain Fabs binding to the BLV-bound NTCP. In the first round, 250 nM of the target was immobilized on streptavidin magnetic beads. Over the course of several rounds of selection, the protein concentration was gradually dropped to 10 nM in the last round. From the second round onwards, the bound phages were eluted using 100 mM glycine, pH 2.7. This harsh elution technique often results in the elution of non-specific and streptavidin binders. To eliminate them, the precipitated phage pool from the second round onwards was negatively selected against streptavidin beads before adding to the target. The pre-cleared phage pool was then used as an input for the selection.

### Single-point phage ELISA

A single-point phage ELISA was used to rapidly screen the binding of the obtained Fab fragments in phage format. The ELISA experiments were also performed at 4 °C in the presence of 1 μM BLV peptide. Colonies of *E. coli* XL1-Blue harboring phagemids from the last round of selection were inoculated directly into 500 μL of 2 × YT broth supplemented with 100 μg/mL ampicillin and M13-KO7 helper phage. The cultures were grown overnight at 37 °C. The experimental wells in the ELISA plates were incubated with 40 nM biotinylated NTCP in ELISA buffer in the presence of 1 μM BLV peptide for 20 min. Only buffer was added to the control wells. Overnight culture supernatants containing Fab phage were diluted 10-fold in ELISA buffer containing 1 μM BLV peptide. The diluted phage supernatants were then transferred to ELISA plates that were pre-incubated with the biotinylated target and washed with ELISA buffer. The ELISA plates were incubated with the phage for another 15 min and then washed with ELISA buffer. The washed ELISA plates were incubated with a 1:1 mixture of mouse anti-M13 monoclonal antibody (GE, 1:5,000 dilution in ELISA buffer) and peroxidase-conjugated goat anti-mouse IgG (Jackson Immunoresearch, 1:5000 dilution in ELISA buffer) for 30 min. The plates were again washed, developed with TMB substrate, and then quenched with 1.0 M HCl, and the absorbance at 450 nm was determined. The background binding of the phage was monitored by the absorbance from the control wells.

### Sequencing, cloning, expression, and purification of Fab fragments

From phage ELISA, positive clones (selected based on a high ratio of ELISA signal of target binding to background) were identified and six unique clones were obtained after DNA sequencing. These six binders were sub-cloned in pRH2.2, an IPTG inducible vector for expression of Fabs in bacteria. *E. coli* C43 (Pro+) cells were transformed with sequence-verified clones of Fab fragments in pRH2.2[66]. The Fabs were expressed and purified according to published protocols[67].

### Multipoint protein ELISA for $EC_{50}$ determination

Multipoint ELISA was performed at 4 °C to estimate the affinity of the Fabs to NTCP in the presence of BLV. 40 nM of target immobilized on a neutravidin-coated ELISA plate was incubated with serial dilutions of the purified Fabs for 20 min. The plates were washed, and the bound NTCP-Fab complexes were incubated with HRP-conjugated Pierce recombinant protein L (Thermofisher, 1:5,000 dilution in ELISA buffer) for 30 min. The plates were again washed, developed with TMB substrate, and quenched with 1.0 M HCl, and absorbance ($A_{450}$) was determined. To determine the affinities, the data were fitted in a dose-response sigmoidal function in GraphPad Prism, and $EC_{50}$ values were calculated.

### Fab-binding nanobody expression and purification

The Fab-binding nanobody[49] construct was created by fusing one N-terminal His-tag and TEV protease cleavage site in a pET26b (+) vector. This plasmid was transformed into *E. coli* BL21 (DE3) cells for expression in a terrific broth (TB) medium. The cells were grown to $OD_{600}$ of 0.8 at 37 °C, and then induced by the addition of 1 mM IPTG and grown for 20 h at 20 °C with shaking. The cells were harvested and the Fab-binding nanobody (Nb) was purified. Periplasmic protein was obtained via osmotic shock by sucrose gradient. The lysate was purified by Ni-NTA chromatography, and the His-tag was removed with TEV protease.

### Cryo-EM sample preparation of NTCP-BLV-Fab3-nanobody complex

First, the BLV peptide was added to the L-MNG-solubilized NTCP solution to a final concentration of 1 μM. Then, the sample was mixed with Fab3 and Fab-binding nanobody at a molar ratio of 1:1:1.2. The mixture was incubated on ice for 1 h and then purified by SEC in buffer containing 25 mM HEPES (pH 7.5), 150 mM NaCl and 0.01% L-MNG. The collected fraction of the NTCP-BLV-Fab3-Nb complex was

concentrated to ~3.6 mg/mL using a 50 kDa molecular weight cut-off centrifugal filter. For grid preparation, 3.5 μL of the sample was applied to glow-discharged Quantifoil R1.2/1.3 carbon/copper 300 mesh grids, and then the grids were plunge-frozen in a liquid ethane-propane mixture cooled by liquid nitrogen, using a Vitrobot Mark IV (Thermo Fisher Scientific) at 4 °C and 100% humidity.

## Cryo-EM data acquisition and processing

Data was collected on a Titan Krios 300 kV (Thermo Fisher Scientific) equipped with a Gatan BioContinuum energy filter and a Gatan K3 detector. Data were acquired with EPU2 (Thermo Fisher Scientific) at a nominal magnification of 130,000× (0.648 Å/pix). The defocus ranged from −0.5 to −2.5 μm, with a total dose of 55 e$^-$/Å$^2$. Data acquisition statistics are presented in Supplementary Table 1 and the data processing pipeline is presented in Supplementary Fig. 3.

The multi-frame movies of the NTCP-BLV-Fab3-nanobody complex were imported into CryoSPARC v4.4.0. All movies were subjected to patch motion correction and CTF estimation using Patch CTF. Several micrographs were used for initial particle selection by blob picker and initial 2D classification. The results from the 2D classification were then used to pick 1,696,776 particles from all micrographs, followed by particle extraction with a pixel size of 0.648 Å/pix. After two rounds of 2D classification, 168,595 particles were used for ab-initio reconstruction to generate three initial models and the best was used for further processing. All particles (663,583) were further subjected to heterogeneous refinement, where after each round the best class was selected for further processing. In total, five subsequent rounds of heterogeneous refinement were performed. The remaining 153,166 particles were then subjected to non-uniform refinement and local refinement, yielding a cryo-EM density map at 3.43 Å resolution. The particles were then subjected to local motion correction, 2D classification that yielded a final set of 128,700 particles, global CTF refinement, local CTF refinement, and another local refinement, yielding a final EM density map at 3.41 Å.

## Model building and refinement

The final cryo-EM map was used for model building in Coot[68]. We built the model for NTCP-BLV-Fab3-nanobody complex based on the previously reported structure of substrate-bound NTCP (PDB ID: 7ZYI). The N-terminus (residues 1-10) and C-terminus (residues 312-349) of NTCP are highly flexible and were not resolved. Model refinements were performed in Phenix[69] with geometric and secondary structure restraints and validated in MolProbity[70]. The BLV peptide showed well-resolved EM density that allowed for de novo model building based on the peptide sequence. The atomic coordinates and geometrical restraint of the ligands (myristoylated glycine) were drawn and generated by ChemDraw (PerkinElmer Informatics, Inc. Version 22.2) and Phenix eLBOW.

## Figure preparation

Graph preparation and statistical analysis were performed in GraphPad 9 (macOS, GraphPad Software, La Jolla California USA, www.graphpad.com) or GraphPad Prism 10 (for Windows). Sequence alignments were generated using Clustal Omega online software[71]. The figures of models and EM density maps were prepared in PyMOL (the PyMOL Molecular Graphics System, Version 2.5, Schrödinger), UCSF Chimera[72], and UCSF ChimeraX[73].

## Reporting summary

Further information on research design is available in the Nature Portfolio Reporting Summary linked to this article.

## Data availability

The electron microscopy density map of the NTCP-BLV-Fab3-Nb complex has been deposited in the Electron Microscopy Data Bank (EMDB) under accession code EMD-19440. The refined model of the complex has been deposited in the Protein Data Bank under accession code 8RQF. Source data are provided with this paper.

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

## Acknowledgements

We thank the Scientific Center for Optical and Electron Microscopy (ScopeM) facility at ETH Zürich for technical support. This research was supported by Swiss National Science Foundation (SNSF) grants 189111 and 214834 to K.P.L.; the Deutsche Forschungsgemeinschaft (DFG) SFB 1021, project number 197785619 (project B8 to J.G. and D.G.); National Institutes of Health grant GM117372 (to A.A.K.). The National Reference Centre for Hepatitis B Viruses and Hepatitis D Viruses at Justus Liebig University Giessen (D.G.) is supported by the German Ministry of Health via the Robert Koch Institute, Berlin, Germany. S.U. received funding from the "Deutsches Zentrum für Infektionsforschung" (DZIF, German Center for Infection Research) – TTU 05.709.

## Author contributions

H.L. and K.P.L. conceived the project. H.L. expressed and purified human NTCP, processed EM data, and built the atomic model with help from K.N. D.Z. and K.N. designed preS1 experiments. S.K. and D.Z. conducted transport assays of NTCP. R.N.I. collected and processed EM data. H.L., K.N., R.N.I., and K.L. performed structural analysis. S.M. and L.R. generated conformational Fabs against NTCP. A.A.K. supervised synthetic antibody generation. D.Z., N.G., and S.K. conducted the SVP and preS1-AX568 binding experiments. R.B.-S. designed and performed NTCP-BLV functional assays and analyzed functional data. J.G. and D.G. contributed to the discussion of the data. S.U. provided the BLV peptide and insight during manuscript preparation. H.L., R.N.I., R.B.-S., K.N., J.G., D.G., and K.P.L. wrote the manuscript with input from the other authors.

## Competing interests

Stephan Urban is the inventor of and holds patents on Bulevirtide, under the patent number WO2019219840A1. The remaining authors declare no competing interests.
