## [Peer Review File · Nature Communications]

Structure of antiviral drug Bulevirtide bound to hepatitis B and D receptor protein NTCPREVIEWERS' COMMENTS

Reviewer #1 (Remarks to the Author):

The manuscript titled “structure of hepatitis B/D antiviral drug bulevirtide bound to its receptor protein NTCP” by Liu and Locher et al reports cryoEM structure of human NTCP bile acid transporter in complex with bulevirtide. Structure determination was facilitated by an antibody fragment Fab3 and a nanobody that binds to Fab3. The study also shows that while transport of bile acid by NTCP is dependent on sodium ions on the extracellular side, binding of preS1 to NTCP is not dependent on the presence of sodium ions. In addition, BLV was shown to inhibit transport activity of NTCP, which is consistent with previous report of similar inhibitory effect of the preS1 peptide. Patient derived subviral particles are shown to bind recombinantly produced NTCP.

The structure of NTCP in complex with BLV is of sufficient resolution to resolve detailed molecular level interactions between the two. The authors used the structure to define the “plug” and the “string” domains of BLV and describe their interactions to NTCP. The N-terminal myristol group of BLV seems to anchor the ligand but is not involved in the molecular recognition. In addition, two bile acid molecules were partially resolved in the structure, and this is consistent with the previous NTCP structure reported by the Locher group. The structure allows for understanding of selective interactions of BLV, and by extension preS1, with NTCP, and provides interpretations for functional impacts of preS1 or BLV on NTCP as well as antiviral effect of a human mutation and species specificity of the viruses.

Overall, the manuscript is well written and has set a high standard for data quality and rigor, and the conclusions are solid. The authors’ analysis of the structure is insightful. I have a minor comment on the two bile acid molecules in the structure. It would be good to use a binding assay to show the stoichiometry of bile acid to NTCP, and the results will likely define the occupancy of the two sites and thus provide hints on the mechanism of transport. However, this is not a demand for additional experiments for the current study.

Reviewer #2 (Remarks to the Author):

In this manuscript, Liu et al. determined the cryo-EM structure of BLV-bound human NTCP. The authors generated FABs using phage-display technology, which helped them to solve the structure. They also presented functional results to demonstrate the role of BLV and the binding of preS1 of HBV SVP and NTCP. This study is a further advance after the authors (and other groups) reported the substrate-bound NTCP structure. Together with the recently reported structure of preS1 peptide-bound NTCP (NSMB

2024), the structures provide insightful information into the interaction between NTCP and HBV S1. Although the similar study, and actually most of structural details, of preS1 peptide-bound NTCP structure has been recently reported, this manuscript provides additional information, e.g., the clear density of myristoyl moiety and an opportunity to compare them directly for potential interactions variations on specific residues and NTCP species. I recommend publication of this manuscript in NC.

Major:

1. Given the similarity and difference between BLV and preS1, the two structures (BLV-bound, and preS1-bound) could be directly compared.

(1) An additional supplementary figure to Fig. 6b showing the interaction around residue 158 in preS1-bound NTCP structure is suggested to supply as mutation Arg158 into Gly158 is conserved in human, and G158R is found abolishing the binding of NTCP to HBV SVP.

(2) NTCP variant S267F was found important and could be explained in the current structure, while residues Q264, T268, and V272 were claimed critical. An additional supplementary figure to Fig. 6c is also suggested for the direct comparison between BLV and PreS1-bound NTCP structures and some related discussions are also suggested.

2. The clear density and location of the myristoyl moiety (Fig. 4d) is interesting. The structure information of this density and their potential role in the interaction between preS1 and NTCP could be commented.

Minor:

1. Why Nano body is used? Does it help to improve the structure determination?

Reviewer #3 (Remarks to the Author):

This study by Hongtao Liu and colleagues demonstrated a detailed structure of Bulevirtide and NTCP, which has been of great medical and scientific interest since this information is critical for designing a novel inhibitor and understanding the molecular mechanism for developing Bulevirtide-resistant viruses. Overall, this is a well-designed and comprehensive study that attracts the attention of both clinical researchers and basic science researchers. Most of my concerns are minor.

Concerns.

1. The authors used Tet-driven 293T cells to induce NTCP expression. I know 293T cells are useful as an in vitro model, but it is much better to investigate the function of NTCP using a hepatocyte-derived cell line such as HepG2 and Huh7 cells since the subcellular localization of NTCP can be cell type dependent (PMID: 11352825) and is modulated by hepatic components such as cholesterol (PMID: 35955590).
2. The authors should spell out all the abbreviations in the first appearance (e.g., “NBD-TC” in Line 103, “ECL1” in Line 158, and “L-MNG” in Line 304).
3. The authors should explain how they determined the incubation time for the inhibition assay in Line 346.
4. Including a catalog number for each reagent is necessary so that other groups can reproduce the findings in this study.
5. In Figure 2b, the authors should show the expression level of NTCP upon Dox induction.
6. The authors should use “mL” instead of “ml” in Line 375 in accordance with other “mL” throughout the manuscript.
7. The S267F variant of NTCP is of interest. The authors can add information about the frequency of this SNP in East Asia populations.
8. The authors can discuss why Cynomolgus monkey NTCP with G158R substitution and S267F variant in East Asia populations were selected during their evolution. Was there selective pressure by Orthohepadnavirus?

Responses to the Reviewers

We would like to thank all three reviewers for their positive remarks. Below are our responses (in blue).

Reviewer #1 (Remarks to the Author):

The manuscript titled “structure of hepatitis B/D antiviral drug bulevirtide bound to its receptor protein NTCP” by Liu and Locher et al reports cryoEM structure of human NTCP bile acid transporter in complex with bulevirtide. Structure determination was facilitated by an antibody fragment Fab3 and a nanobody that binds to Fab3. The study also shows that while transport of bile acid by NTCP is dependent on sodium ions on the extracellular side, binding of preS1 to NTCP is not dependent on the presence of sodium ions. In addition, BLV was shown to inhibit transport activity of NTCP, which is consistent with previous report of similar inhibitory effect of the preS1 peptide. Patient derived subviral particles are shown to bind recombinantly produced NTCP.

The structure of NTCP in complex with BLV is of sufficient resolution to resolve detailed molecular level interactions between the two. The authors used the structure to define the “plug” and the “string” domains of BLV and describe their interactions to NTCP. The N-terminal myristol group of BLV seems to anchor the ligand but is not involved in the molecular recognition. In addition, two bile acid molecules were partially resolved in the structure, and this is consistent with the previous NTCP structure reported by the Locher group. The structure allows for understanding of selective interactions of BLV, and by extension preS1, with NTCP, and provides interpretations for functional impacts of preS1 or BLV on NTCP as well as antiviral effect of a human mutation and species specificity of the viruses.

Overall, the manuscript is well written and has set a high standard for data quality and rigor, and the conclusions are solid. The authors’ analysis of the structure is insightful. I have a minor comment on the two bile acid molecules in the structure. It would be good to use a binding assay to show the stoichiometry of bile acid to NTCP, and the results will likely define the occupancy of the two sites and thus provide hints on the mechanism of transport. However, this is not a demand for additional experiments for the current study.

We thank the reviewer for the very kind evaluation. We would like to note that in our BLV-bound NTCP structure, we do not have any bile acid molecules. In Fig. 5, where bile acids are shown, we superimposed our current structure with the structure from our previous study (PDBID 7ZYI). The previous structure revealed two bile acid molecules; therefore our Fig. 5 shows the incompatibility between binding of BLV and substrate molecules. We also appreciate the comment on the stoichiometry of bile salt molecules binding to NTCP. We will implement the idea in future mechanistic studies.

Reviewer #2 (Remarks to the Author):

In this manuscript, Liu et al. determined the cryo-EM structure of BLV-bound human NTCP. The authors generated FABs using phage-display technology, which helped them to solve the structure. They also presented functional results to demonstrate the role of BLV and the binding of preS1 of HBV SVP and NTCP. This study is a further advance after the authors (and other groups) reported the substrate-bound NTCP structure. Together with the recently reported structure of preS1 peptide-bound NTCP (NSMB 2024), the structures provide insightful information into the interaction between NTCP and HBV S1. Although the similar study, and actually most of structural details, of preS1 peptide-bound NTCP structure has been recently reported, this manuscript provides additional information, e.g., the clear density of myristoyl moiety and an opportunity to compare them directly for potential interactions variations on specific residues and NTCP species. I recommend publication of this manuscript in NC.

We thank the reviewer for the positive feedback.

Major:

1. Given the similarity and difference between BLV and preS1, the two structures (BLV-bound, and preS1-bound) could be directly compared.

(1) An additional supplementary figure to Fig. 6b showing the interaction around residue 158 in preS1-bound NTCP structure is suggested to supply as mutation Arg158 into Gly158 is conserved in human, and G158R is found abolishing the binding of NTCP to HBV SVP. (2) NTCP variant S267F was found important and could be explained in the current structure, while residues Q264, T268, and V272 were claimed critical. An additional supplementary figure to Fig. 6c is also suggested for the direct comparison between BLV and PreS1-bound NTCP structures and some related discussions are also suggested.

We acknowledge the similarity between the BLV-bound and preS1-bound NTCP structures. We had already done the comparison of the two structures in Supplementary Fig. 7. As shown in Fig. 6, both G158 and S267 residues interact with the plug region of the peptide. The plug region is highly conserved among preS1 peptides and BLV (Fig. 4c) and therefore adopts the same conformation in the translocation tunnel of NTCP. It is challenging to show these interactions in a 2D figure, and we therefore hope that interested readers will refer to the PDB model for further analysis.

2. The clear density and location of the myristoyl moiety (Fig. 4d) is interesting. The structure information of this density and their potential role in the interaction between preS1 and NTCP could be commented.

Our original text reads “The myristoyl group of BLV interacts with the surface of TM4 (Phe128, Leu131, and Met133) and TM5 (Tyr156) of NTCP and is exposed to lipids from the outer leaflet of the basolateral hepatocyte membrane.” We believe that any further comments regarding interaction between the myristoyl moiety of preS1 and NTCP would be too speculative.

Minor:

1. Why Nano body is used? Does it help to improve the structure determination?

The use of a nanobody in our structural studies is twofold – (1) to stabilize the hinge of the Fab and therefore reduce Fab flexibility and (2) as an asymmetrical marker.

Reviewer #3 (Remarks to the Author):

This study by Hongtao Liu and colleagues demonstrated a detailed structure of Bulevirtide and NTCP, which has been of great medical and scientific interest since this information is critical for designing a novel inhibitor and understanding the molecular mechanism for developing Bulevirtide-resistant viruses. Overall, this is a well-designed and comprehensive study that attracts the attention of both clinical researchers and basic science researchers. Most of my concerns are minor.

We thank the reviewer for the very kind evaluation.

Concerns.

1. The authors used Tet-driven 293T cells to induce NTCP expression. I know 293T cells are useful as an in vitro model, but it is much better to investigate the function of NTCP using a hepatocyte-derived cell line such as HepG2 and Huh7 cells since the subcellular localization of NTCP can be cell type dependent (PMID: 11352825) and is modulated by hepatic components such as cholesterol (PMID: 35955590).

In the past we reported a comparison between PHH, NTCP-HEK293 and NTCP-HepG2 cells (König et al. 2014, PMID: 24845614) and found that bile acid transport, preS1 binding, and competition of preS1 and bile acids in their binding to NTCP can be concurrently shown in all three cell systems. We do acknowledge that for in vitro infection experiments the cell type would make a difference and we will keep this in mind for future studies.

2. The authors should spell out all the abbreviations in the first appearance (e.g., “NBD-TC” in Line 103, “ECL1” in Line 158, and “L-MNG” in Line 304).

We have checked the text to make sure that all abbreviations are defined.

3. The authors should explain how they determined the incubation time for the inhibition assay in Line 346.

The inhibitor preincubation time was set to 5 min, to be consistent with previously published data (Kirstgen et al. 2020, PMID:33303817). We included this information in the main text and added the corresponding reference.

4. Including a catalog number for each reagent is necessary so that other groups can reproduce the findings in this study.

We have added catalogue numbers to reagents where necessary.

5. In Figure 2b, the authors should show the expression level of NTCP upon Dox induction.

Expression levels of NTCP were determined using qPCR, and as such NTCP expression was detectable in HEK-NTCP cells and not detectable in HEK-293 cells. We have amended the figure caption of Figure 2 to include how expression levels of NTCP were validated.

6. The authors should use “mL” instead of “ml” in Line 375 in accordance with other “mL” throughout the manuscript.

We have corrected the inconsistency.

7. The S267F variant of NTCP is of interest. The authors can add information about the frequency of this SNP in East Asia populations.

We have reworded the text as follows and added the corresponding reference:

“Another noteworthy NTCP site is the nonsynonymous human genetic variant S267F (c.800C>T) of NTCP, which has been identified exclusively among East Asian populations, with a minor allele frequency ranging from 2% to 10%, while it remains absent in African and European populations.”

8. The authors can discuss why Cynomolgus monkey NTCP with G158R substitution and S267F variant in East Asia populations were selected during their evolution. Was there selective pressure by Orthohepadnavirus?

The G158R substitution present among all Old World monkeys, including Cynomolgus macaques, has been reported to be under positive selection to prevent HBV-infection and HBV-related disease progression in those monkeys [1,2]. A similar positive selection factor might also be relevant for the S267F variant. The S267F variant is associated with a reduced HBV infection risk, but does not provide full protection against infection with HBV. It shows however a decreased risk for liver cirrhosis in chronic HBV-infected persons [3].

[1] Müller, S. F., König, A., Döring, B., Glebe, D. & Geyer, J. Characterisation of the hepatitis B virus cross-species transmission pattern via Na⁺/taurocholate co-transporting polypeptides from 11 New World and Old World primate species. *PLoS One* 13, e0199200 (2018). <https://doi.org/10.1371/journal.pone.0199200>

[2] Takeuchi JS, Fukano K, Iwamoto M, Tsukuda S, Suzuki R, Aizaki H, Muramatsu M, Wakita T, Sureau C, Watashi K 2019. A Single Adaptive Mutation in Sodium Taurocholate Cotransporting Polypeptide Induced by Hepadnaviruses Determines Virus Species Specificity. *J Virol* 93:10.1128/jvi.01432-18. <https://doi.org/10.1128/jvi.01432-18>

[3] Ping An, Zheng Zeng, Cheryl A Winkler, The Loss-of-Function S267F Variant in HBV Receptor NTCP Reduces Human Risk for HBV Infection and Disease Progression, *The Journal of Infectious Diseases*, Volume 218, Issue 9, 1 November 2018, Pages 1404–1410, <https://doi.org/10.1093/infdis/jiy355>